# Patient Experiences of Community Pharmacy Medication Supply and Medicines Reconciliation at Hospital Discharge: A Pilot Qualitative Study

**DOI:** 10.3390/pharmacy12020066

**Published:** 2024-04-10

**Authors:** Rhona Mundell, Derek Jamieson, Gwen Shaw, Anne Thomson, Paul Forsyth

**Affiliations:** 1Pharmacy, Glasgow Royal Infirmary, NHS Greater Glasgow & Clyde, Glasgow G4 0SF, UK; rhona.mundell@ggc.scot.nhs.uk; 2Pharmacy, Glasgow City Health & Social Care Partnership (North East Locality), NHS Greater Glasgow & Clyde, Glasgow G1 1LH, UK; derek.jamieson@ggc.scot.nhs.uk; 3Pharmacy, Clarkston Court, NHS Greater Glasgow & Clyde, Glasgow G76 7AT, UK; gwen.shaw@ggc.scot.nhs.uk (G.S.); anne.thomson2@ggc.scot.nhs.uk (A.T.)

**Keywords:** hospital pharmacy, community pharmacy, medication, qualitative research

## Abstract

(1) Background: As part of the Scottish Government’s five-year recovery plan to address the backlog in NHS care following the COVID-19 pandemic, community pharmacies in Scotland are planned to provide a Hospital Discharge Medicines Supply and Medicines Reconciliation Service. We aimed to qualitatively explore patients’ experiences with this new service. (2) Method: Adult patients (≥18 years age) who consented to participate in the Community Pharmacy Hospital Discharge and Medicines Reconciliation Service were invited for an interview within 21 days of discharge from hospital. Qualitative, one-to-one, semi-structured patient interviews were conducted by telephone and audio-recorded using Microsoft Teams^®^. The interview audio recordings were transcribed verbatim and underwent thematic analysis. (3) Results: Twelve patients were interviewed, evenly split by sex and with a median age of 62 years (range 36 to 88 years). Our analysis generated main five themes: patient engagement, stakeholder communication, practical factors, human factors, and comparative experiences. Many of these were interdependent. (4) Conclusions: Patients appreciated that the service ensured a quicker discharge from hospital. Good stakeholder communication, practical factors (including choice, location, and the realities of obtaining their medication from the community pharmacy), and a pre-existing and trusted relationship in their usual community pharmacy were the key factors that regulated the patient experience. Generally, patients were positive about the introduction of this new service. However, the lack of a previous relationship or trust with a community pharmacy, and previous experiences with medication supply problems were factors which had the potential to negatively impact patient experiences.

## 1. Introduction

The Scottish Government’s strategy, Achieving Excellence in Pharmaceutical Care, set out the need for transformational change across all areas of pharmacy practice to ensure safe, effective, and person-centred pharmaceutical care, and the safer use of medicines [1]. The strategy proposed that an aspirational future model would utilise cross-sector pharmacy resources and infrastructure in an integrated manner during the hospital discharge process to improve efficiency, enhance quality of care, and improve health outcomes [1].

Delays in hospital discharge have been an issue in the National Health Service (NHS) for many years. Such delays contribute to bed pressures and hamper patient flow. In England, the National Audit Office reported that 2.2 million bed days could be attributed to delays in discharge in 1998/99, costing NHS England the equivalent of GBP 1 million a day [2]. The annual estimated cost of delayed discharges in NHS Scotland was GBP 142 million in 2020 [3]. Across NHS Scotland in 2021/22, delayed discharges accounted for more than half a million additional days spent in hospital [4]. This problem is growing over time with a 4% increase in delayed discharges between 2018/19 and 2019/20. Delays can occur for a variety of reasons, but a recognised contributing factor can be the time taken to prescribe, dispense, and deliver discharge medicines to the patient [2].

It is recognised that community pharmacy plays an important role in the provision of pharmaceutical care, providing highly accessible services. Medication review, by a patient’s local and familiar community pharmacy team, could also bring key safety improvements, ensuring that intended changes are picked up, screened, and discussed with the patient, reducing medicines-related adverse effects [5]. Following the COVID-19 pandemic, the Scottish Government’s five-year recovery plan, to address the backlog in NHS care, proposed that community pharmacies in Scotland would provide a hospital discharge and medicines reconciliation service to speed up the discharge process and ensure the safer use of medicines [6].

A quality improvement programme, conducted in Glasgow Royal Infirmary in 2020, investigated whether a new Community Pharmacy Hospital Discharge Medicines Supply Service was quicker than the standard hospital discharge process. It found that the supply of medication on hospital discharge by community pharmacies resulted in a median time saved of 142 min per patient, across three Plan–Do–Study–Act cycles in a total of 335 patients in one large inner-city teaching hospital. It concluded that the expansion of this service had the potential to deliver a transformational change in bed management and patient flow within the hospital [7]. Qualitative research of this new pathway is needed, however, to investigate lived patient experiences. We also wanted to test the addition of medicines reconciliation. Therefore, the aim of this new study was to qualitatively explore patients’ experience of the new Community Pharmacy Hospital Discharge and Medicines Reconciliation Service.

## 2. Materials and Methods

*Setting*: This study was conducted in Glasgow Royal Infirmary, a large teaching hospital within NHS Greater Glasgow & Clyde (NHSGGC). NHSGGC is the biggest regional health board in Scotland, providing healthcare to a population of 1.3 million people [8]. Overall, 34% of the most socioeconomically deprived areas in Scotland are within this health board [9]. Glasgow Royal Infirmary is located in an inner-city environment with high levels of health and social inequalities. It has around 1000 beds and is home to some 5000 members of staff, including 41 pharmacists [10].

*Service Model, including Inclusion/Exclusion Criteria*: This service model has already been described in detail [7]. But, in summary, community pharmacies from the NHSGGC geographic boundaries near Glasgow Royal Infirmary were invited to participate in the Community Pharmacy Hospital Discharge and Medicines Reconciliation Service. All adult patients within those regions, who required a supply of medication at hospital discharge, were screened for inclusion. To participate in the service, patients must have been reviewed by a pharmacist during their hospital admission to ensure the discharge medication was clinically appropriate. In addition, patients must have had the required discharge medication prescribed during or before admission.

Patients were excluded if they had significant cognitive impairment, terminal illness, or significant social problems and/or severe mental health comorbidities. Any patient requiring the supply of a controlled drug from schedules 1–4 was also excluded due to restrictions in the law allowing community pharmacies to supply patients medicines without a prescription. In addition, patients were excluded from the service if the community pharmacy was beyond the included health authority regions; did not agree to participate; did not stock the required medication; or if the discharge occurred outside normal working hours.

The hospital healthcare professional (HCP) team established an agreed discharge plan through discussions with the patient and the community pharmacy. Patients were asked about an appropriate choice of community pharmacy and method of obtaining the discharge medication (collection or delivery). The community pharmacy was contacted by telephone to advise them that a patient being discharged was suitable for the service. Discussion with the community pharmacy confirmed whether they were able to (a) access the electronic Immediate Discharge Letter (eIDL) on a shared IT medical record platform, (b) supply the discharge medication to the patient using the agreed method, and (c) complete the medicines reconciliation. When the patient/carer collected the medication, the community pharmacy would discuss or arrange to discuss any new or changed medication within 5 days of discharge.

*Sampling*: Our convenience sampling strategy aimed to recruits a sample of 12 participants between May 2022 and October 2022. This sample size was estimated to achieve data saturation [11]. No formal saturation checking criteria was set a priori.

*Study Participant Recruitment*: Patient information leaflets and consent forms were issued to patients for this qualitative study. No incentives (e.g., financial) were offered for participation. Completed consent forms were returned to the lead author and participants were interviewed by the lead author within 21 days of discharge. Interviews continued until the sample size was reached.

*Data management*: Prior to the interview, a semi-structured interview was developed incorporating four questions to qualitatively explore the patients’ experience of the service—see Appendix A. Questions were structured around the broad concepts of Normalisation Process Theory [12]. Each patient was contacted and invited to participate in a recorded interview using Microsoft Teams^®^. At the interview, each participant reconfirmed their consent and was informed of the aims of the study. Demographic characteristics were collected after the recorded interview was complete. All interviews lasted between 10 and 20 min and were audio recorded; these interviews were transcribed verbatim and anonymised (RM). These transcripts were then accuracy checked by an independent staff member who did not take part in the study. All electronic data were stored on encrypted and password-protected NHS computers. After transcription and validation, all recordings were deleted.

*Data analysis*: The lead author (RM) was a female pharmacist with 32 years of experience working in hospital pharmacy. The lead author coded all interviews. Other coders included a male pharmacist with 20 years’ experience in primary care (PF), a female pharmacist with 32 years’ experience in hospital pharmacy (GS), and a female pharmacist with 24 years’ experience across both hospital and primary care pharmacy (AT), and they independently coded nine interviews (PF × 3, AT × 3, and GS × 3). Transcribed data then underwent thematic analysis over six phases [13,14]. An inductive approach was used to divide the data into meaningful categories and descriptors (i.e., generating initial codes). A meeting between coders was then arranged to discuss the thematic analysis of the data. A preliminary coding scheme was developed and applied across all nine interviews that had been conducted. All the transcribed interviews were then reviewed again with comparisons made within and between interviews (RM). Patterns of this coded data were organized into broader concepts which linked them (i.e., themes) [13,14,15]. Refinements of themes were discussed (RM and PF). Each theme had a clear narrative that was relevant to the aim of this study. The agreed themes and subthemes were then critiqued and refined further (AT and GS). Three further interviews were conducted, reaching a sample size of 12 participants. These were then coded and compared against the thematic structure and recruitment finished at that point, and saturation was judged to have been achieved. The reporting of this study follows the consolidated criteria for reporting qualitative studies (COREQ) guidelines [16].

*Ethics approval*: The local NHS research ethics department advised that ethical review was not required on the basis that this study was a service evaluation. Representatives from the local NHS governance department gave advice to the study-steering group about data handling.

## 3. Results

Forty patients were contacted to participate. Twelve patients consented and were interviewed; their characteristics are presented in Table 1. The patients were evenly split by sex with a median age of 62 years (range 36 to 88 years). The majority of patients (*n* = 7) lived within the most socio-economically deprived quintile [9]. Patients were discharged from a variety of clinical specialist ward types.

The analysis generated five main themes: patient engagement (sub-themes motivation, and perceived reason/need for new discharge service), stakeholder communication (sub-themes hospital HCPs and community HCPs), practical factors (sub-themes community pharmacy choice, community pharmacy location, and obtaining medication), human factors (sub-themes community pharmacy relationship and professional trust), and comparative experiences. A narrative text of each theme and sub-theme is detailed below, including illustrative quotes.

### 3.1. Patient Engagement

#### 3.1.1. Motivation

Generally speaking, patients did not want to stay in hospital any longer than necessary, and, when well enough to do so, were keen to get home. Most patients expressed this sentiment, and this was a key facilitator in the patients’ motivation to participate in the new service.

“I think everybody is just desperate to get out of hospital”Patient 2 (Female, 79 years old)

#### 3.1.2. Perceived Reason/Need for New Discharge Service

Aligned to their personal motivation, patients typically understood that the new service was designed to be quicker and more efficient.

“I was told it was for quickness”Patient 6 (Male, 56 years old)

Numerous patients had personal experience with previous occasions of long delays in discharge whilst waiting for their medication to be supplied by the hospital pharmacy, or appreciated this problem through witnessing the inefficiency of other patients/relatives going through the traditional supply model.

“A patient missed her transport waiting on medication”Patient 9 (Female, 79 years old)—describing witnessed experiences of hospital discharge supply model

“You can be waiting 3, 4 [h] maybe longer for the prescription coming up”Patient 12 (Male, 59 years old)—describing previous experiences of hospital discharge supply model

### 3.2. Stakeholder Communication

#### 3.2.1. Hospital HCP

In the majority of cases, appropriate communication from the hospital team facilitated a smooth community pharmacy supply. This included communication with the patient to understand which community pharmacy they normally used and good communication with the chosen community pharmacy itself.

“a pharmacy person had came and they had checked my medication and they had got in touch [name of usual pharmacy] and arranged everything for me”Patient 7 (Female, 71 years old)

Effective communication could be challenging; there were numerous distractions within a busy ward environment and although ready to be discharged from hospital, patients were still recovering.

“She didn’t get a lot of time because obviously people are there to take your bloods and stuff like that so she was kinda trying to do the best she could to explain”Patient 4 (Female, 36 years old)

There was a risk that the patient may not understand or may forget the instructions given, particularly when it is unfamiliar.

“I cant mind too much what they said I was that excited about getting out ye know”Patient 8 (Male, 75 years old)

#### 3.2.2. Community HCP

Many patients described being contacted by the community pharmacy to advise them when their prescription would be ready to collect, and some patients described having the medication delivered directly to their home.

“the pharmacy phoned my husband and said it would be ready for two o’clock”Patient 4 (Female, 36 years old)

However, three patients’ expectations of the service were not met when the community pharmacy had not accessed the patient’s discharge letter in time and did not seem to expect them, or did not have their medication ready for collection when the patient arrived.

“I don’t think they’d emailed the actual medication I was supposed to get. They gave me a sheet of paper with all my medication on it so the pharmacy was able to deal with it that way”Patient 1 (Male, 53 years old)

### 3.3. Practical Factors

#### 3.3.1. Community Pharmacy Choice

Almost all patients who participated in the study had “their own” community pharmacy. This made it easier to plan their discharge as the patient and community pharmacy were familiar with one another.

“I got it from my ain chemist, I got ma ain chemist to deliver it”Patient 3 (Male, 83 years old)

“my ain pharmacy gave me it. You know, my ain, one a get ma own tablets fae, They gave me it”Patient 9 (female, 79 years old)

#### 3.3.2. Community Pharmacy Location

Ideally, the community pharmacy should be convenient for the patient/carer to use and for most patients this is close to their home. An agreed community pharmacy was chosen for one patient in the study who did not have ‘his own’ community pharmacy. However, the patient experienced an unacceptable travel burden and subsequently made arrangements to collect his prescription from a community pharmacy closer to his home.

“With me being able to just go down the street now and get the medication is a big difference. It’s a lot easier for me”Patient 1 (Male, 53 years old)—his own choice of CP

#### 3.3.3. Obtaining Medication

Some patients’ discharge medication was delivered to their home, but most prescriptions were collected in person and frequently this involved the help of a family member.

“ma granddaughter picked it up and a got it right away”Patient 11 (Female, 88 years old)

The collection of medication may be problematic for patients who are still recovering from acute illness, and some patients expressed an understanding of the potential burden of collection on family members

“Ah felt sorry for ma son running about”Patient 9 (female, 79 years old)

### 3.4. Human Factors

#### 3.4.1. Community Pharmacy Relationship

A good pre-existing relationship with the community pharmacy was a key enabler. Many patients spoke positively of the staff and the service provided by their pharmacy.

“It actually is a good service. You know what, I’ve been a couple of times now to it and you know they are good, good people”Patient 1 (Male, 53 years old)

One patient preferred collecting his prescription over having it delivered, as he valued the personal contact.

#### 3.4.2. Professional Trust

Many patients described receiving helpful information and advice about their medication from their community pharmacy. This could vary from practical help on supply issues to advice about specific medication reconciliation issues.

“She took the aspirin off me. I’ve got tae watch if ah bleed”Patient 9 (Female, 79 years old)

“they just change some without telling you really and you wonder, you wonder why. But I was told why, so that was good”Patient 8 (Male, 75 years old)

One patient was happy for appropriate information to be shared between the hospital and his community pharmacy to ensure he was treated as a patient, not a customer.

“They’ve also got a direct link with the hospital so when I get discharged for anything they’ll get a copy of my medical report which I’m quite happy with I don’t mind that, you know, cause then they actually know me rather than just a customer, like a patient if you like.”Patient 12 (Male, 59 years old)

However, one patient did not like talking about her health in the community pharmacy, implying concerns about privacy and trust. It was also noted that some pharmacies may be more like a business than a healthcare service.

“I’m not one for having a lot of conversations with people in pharmacies and things like that”Patient 5 (Female, 64 years old)

### 3.5. Comparative Experience

Most patients found the new service quicker and more convenient.

“I actually couldn’t believe how painless it was”Patient 4 (Female, 36 years old)

“I thought it was a lot quicker”Patient 9 (Female, 79 years old)

For those patients who had an established relationship with their community pharmacy, the familiarity of service was greatly valued.

“its easiest for the, for patients…They know that they’ll receive it, you know, by their normal methods”Patient 7 (Female, 71 years old)

However, the patients’ experience of the new service was detrimentally affected if they had to wait at the community pharmacy for their prescription, if stock issues necessitated return visit(s) to the community pharmacy, or if the agreed pharmacy was inappropriate.

“I had to go back down the next day to collect other medication because they didn’t have it in stock”)Patient 1 (Male, 53 years old

Most patients expressed a preference for the new service and responded positively towards introducing the new service more widely, although it was recognised that it might not be suitable for everyone.

“better from the community pharmacy, a lot better. I think that would be better for everyone”Patient 9 (Female, 79 years old)

However, a few were ambivalent, and two patients preferred the current hospital discharge service. Whilst acknowledging that it could involve a lengthy wait, they felt it was more convenient and trustworthy; you were certain to be given the correct medication (incorporating any changes).

“I know ye have to wait on the hospital one for to come but still I think it was better that way”Patient 10 (Male, 79 years old)

## 4. Discussion

Patients’ experience was dependent on a number of interrelated factors. These included Patient engagement, stakeholder communication, practical factors, and human factors. The majority of patients expressed positive comparative experiences and a preference for the new service. However, it is also important to consider detrimental experiential factors as these highlight potential areas for service improvement.

All the patients who participated in the study described being discharged from hospital more quickly. This was a key reason for patient engagement. It is well recognised that in the current hospital discharge supply model, considerable delays can be experienced by patients waiting to obtain their medicines [17,18]. Many patients find this a frustrating and anxious time [17,18,19]. In addition, this discharge delay can temporarily block beds [20]. Timely discharge from hospital is an important indicator of quality, person-centred care, and ensures that patients get back to their home or community environment as soon as possible [21,22,23]. The transformation of hospital discharge planning and supply of medicines is a key priority of the NHSGGC [24]. In other countries, such as the Netherlands, it is standard practice to send discharge prescriptions directly from the hospital to the community pharmacy where the patient is registered [25,26].

Good communication between stakeholders is essential to plan discharge. Hospital discharge is a complex process and the need for better co-ordination, communication, and information-sharing between hospital- and community-based HCPs is a recognised challenge [20,27]. Transitions of care (ToC) are a high-risk period for medication-related harm, due to the volume and complexity of medication changes [28,29]. Poor communication between HCPs continues to be a common contributing factor to medication errors [29,30]. Preventable harm from medicines is thought to cost the NHS anywhere between GBP 1 billion and GBP 2.5 billion annually [20]. The prevention of medication errors by improving ToC has become a high priority worldwide, and numerous interventions have been designed and delivered to try and address this issue [20,30,31,32]. Medication reconciliation, the formal process of obtaining a complete and accurate list of each patient’s current medication across ToC, has been widely implemented and endorsed to prevent medication discrepancies [25,33]. Patient discharge information should be shared with their nominated community pharmacy, and medicines reconciliation in primary care should be carried out as soon as is practically possible, ideally within one week, to enhance patient safety relating to medicines [34,35]. Despite these guidelines, the communication of discharge summaries to community pharmacy remains inconsistent, incomplete, and lacking in timeliness, and the potential for community pharmacy involvement in the discharge process is underutilised [36]. Our study confirms that communication processes between hospital and community pharmacies must be strengthened to ensure the dependable transfer of patient discharge information to their community pharmacy [37,38].

The majority of patients found using “their own” community pharmacy to obtain their discharge medicines natural and familiar, and many patients described receiving help and advice about their medicines from their community pharmacy. Patients can feel overwhelmed by medication changes which have occurred during hospitalisation and feel that healthcare providers do not always notice the impact this can have and the need for support to cope with these changes [18]. Discharge counselling has been recommended to support adherence; however, providing medicines information at discharge is not always ideal as patients are often distracted, overloaded with information, and eager to leave for home, and therefore pay less attention to medication instructions [33,39,40]. Community pharmacists are well positioned to provide medicines information after discharge [39,40]. Patients prefer face-to-face consultations with someone who understands their personal issues, preferences, and needs and can empathise with them [18]. However, patients’ lack of mobility and reluctance to seek community pharmacists’ advice about their medication are known barriers [40,41,42]. Communication breakdowns at ToC can have an impact on patients’ trust in their HCP; they fear medication errors could occur [18]. It has also been reported that patients view their GP as their primary source of information about their medicines and do not understand the value of a community pharmacy medication review [37,40,41,42,43]. The importance of long-term relationships between patients and HCPs in primary care is recognised [22]. The strength of the relationship between pharmacist and patient has been identified as a key factor in a patient’s decision to access community pharmacy services [40]. It has been recommended that to inform the public about the added value of a consultation with a pharmacist and to improve the uptake of pharmacy services, awareness campaigns should focus on encouraging patients to build relationships with their community pharmacists rather than targeting and marketing specific services that can be offered [37,40]. Continuity of care with community pharmacy has also been shown to improve medication safety [44].

Two patients preferred the current hospital discharge supply model, despite acknowledging that it could involve a lengthy wait. They believed the hospital pharmacy would dispense their medication without any issues, incorporating any changes, and they regarded it as more convenient as they were already in hospital. They recollected their community pharmacy not stocking items, which necessitated a return visit, and the hospital wait was also compared to the potential need to wait at the community pharmacy. Reviewing the discharge process is important, but in doing so it is valuable to acknowledge that speed of discharge may not be as important as the patient developing a trusted professional relationship with the community pharmacy.

### 4.1. Strengths and Weaknesses

This novel study used an appropriate qualitative methodology. The study identified potential barriers and facilitators, providing valuable insights into the effectiveness and challenges of this process, from a patient perspective. The study’s findings may inform interventions to improve the management of patients discharged from hospital and their subsequent management in the community. The study’s use of COREQ guidelines in reporting its findings ensures its findings can be easily replicated and implemented in practice.

There are some limitations to this study. Firstly, no formal saturation stopping criteria were set a priori (i.e., we used judgement, rather than a predefined rule), which may have led to a lack of full saturation. The study included mainly elderly patients, who may have different experiences and preferences compared to younger patients. Additionally, most participants had their own community pharmacy, which may limit the generalizability of the findings to patients who do not have an established relationship with a community pharmacy. There is also potential for recruitment bias if patients with a bad experience are self-excluded from the study. At patient recruitment, patients were advised the new service was intended to be quicker, and this may have biased qualitative results. The limitations of the study should be taken into account when interpreting the study’s finding and considering their applicability to other patient populations or settings.

### 4.2. Further Research

Future studies could investigate the experience of staff members and/or younger patients who do not have an existent relationship with a community pharmacy. A further study with a larger sample and a pre-specified data saturation rule may also give more certainty to the result.

## 5. Conclusions

In this pilot study, patients appreciated that a service where a community pharmacy supplied their discharge medication ensured a quicker discharge from hospital. Thematic analysis also showed that good stakeholder communication, practical factors (including choice of pharmacy, location of pharmacy, and the practicalities of obtaining the supply from a community pharmacy), and a pre-existing relationship and trust in their normal community pharmacy were the key factors that regulated the patient experience. On the whole, patients were positive about the introduction of this new service. However, a lack of previous relationship or trust with a community pharmacy, and previous experiences with medication supply problems were factors which had the potential to negatively impact patient experiences. Further research with a larger sample size and a pre-specified data saturation rule is needed to confirm these results.

## Figures and Tables

**Table 1 pharmacy-12-00066-t001:** Characteristics of participating patients.

Characteristics	Number of Patients (*n* = 12)
Median age (range), years	62 (36–88)
Female sex	6 (50%)
Socio-economic Deprivation Quintile *	
1 (most deprived)	7 (58.3%)
2	1 (8.3%)
3	3 (25.0%)
4	1 (8.3%)
5 (most affluent)	0 (0%)
Clinical speciality of ward patient discharged from	
Respiratory	5 (41.6%)
Endocrinology/diabetes	2 (16.6%)
Gastroenterology	2 (16.6%)
General medicine	2 (16.6%)
Cardiology	1 (8.3%)

* = Measured by the Scottish Index of Multiple Deprivation score, based on the post code of the patient’s home address.

## Data Availability

Raw data from the study are unavailable due to privacy.

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
