# Peer review of "Patient Experiences of Community Pharmacy Medication Supply and Medicines Reconciliation at Hospital Discharge: A Pilot Qualitative Study"

_pharmacy, 2024, doi:10.3390/pharmacy12020066_

Round 1

Reviewer 1 Report

Comments and Suggestions for Authors

Thank you for the opportunity to review this interesting, well written submission. I have only a few small comments to make.

I understand the rationale behind publishing the qualitative data separate from the overall analysis but it would assist the reader to have a little more detail about the size of the initiative and reported benefits (from ref 7). In the acknowledged absence of a saturation analysis, and only 12 participants, it was difficult to get a sense of the impact of the service, and whether the data presented in this paper was reflective of the larger analysis. I would also have liked to have seen some more data in the results, some of the subthemes were limited to a single quote and it is hard for the reader to determine how common the sentiments were among the other participants.

How many participants did you approach to get 12 who consented?

In the section on data collection and handling you refer to 5 questions in the semi-structured interview but Appendix A seems to only have 4 questions?

In your data analysis (line 131) you refer to 10 coded interviews but have only allocate 9 to the authors (PF, AT, GS 3 each)

Reference 38 seems to be cited after reference 39

Reference 29 seems to be missing a title?

Author Response

Reviewer 1: I understand the rationale behind publishing the qualitative data separate from the overall analysis but it would assist the reader to have a little more detail about the size of the initiative and reported benefits (from ref 7).

Reply: Thank you. We have now slightly expanded the introduction to read ‘A quality improvement programme, conducted in Glasgow Royal Infirmary in 2020, investigated whether a new Community Pharmacy Hospital Discharge Medicines Supply Service was quicker than the standard hospital discharge process. It found that the supply of medication on hospital discharge by community pharmacies resulted in a median time saving of 142 minutes per patient, across three Plan-Do-Study-Act cycles in a total of 335 patients in one large inner-city teaching hospital.’

Reviewer 1: In the acknowledged absence of a saturation analysis, and only 12 participants, it was difficult to get a sense of the impact of the service, and whether the data presented in this paper was reflective of the larger analysis. I would also have liked to have seen some more data in the results, some of the subthemes were limited to a single quote and it is hard for the reader to determine how common the sentiments were among the other participants.

Reply: Thank you. We have added in a few additional quotes throughout, to give the reader a broader understanding of the themes or sub-themes.

Reviewer 1: How many participants did you approach to get 12 who consented?

Reply: Thank you. We have adapted the first line of the results section to now read ‘40 patients were contacted to participate. 12 patients consented and were interviewed

Reviewer 1: In the section on data collection and handling you refer to 5 questions in the semi-structured interview but Appendix A seems to only have 4 questions?

Reply: Thank you for highlighting this typo. We have now corrected this to ‘four questions’ in the main text.

Reviewer 1: In your data analysis (line 131) you refer to 10 coded interviews but have only allocate 9 to the authors (PF, AT, GS 3 each)

Reply: Thank you for highlighting. We have updated this to ‘nine’ and we have corrected this paragraph in a few places, so that it is consistent.

Reviewer 1: Reference 38 seems to be cited after reference 39

Reply: Thank you. We had missed citing 38 early in the paper and have now added this in.

Reviewer 1: Reference 29 seems to be missing a title?

Reply: Thank you for highlighting this typo. We have now corrected this and added the missing title ‘Refer-to-pharmacy: a qualitative study exploring the implementation of an electronic transfer of care initiative to improve medicines optimisation following hospital discharge’

Reviewer 2 Report

Comments and Suggestions for Authors

This is an excellent paper, extremely relevant and well written. 

Three  comments 

Lines 110 and 11 - Explain how you arrived at saturation  at 12 - relative to the paper by Hennink et al 

Lines 116 and 117 -Explain the development of the questions relating to the terms in brackets in the interview schedule e.g question 1 (coherence) 

How was the demographic information in table 1 obtained - not included in the interview. 

Author Response

Reviewer 2: Lines 110 and 11 - Explain how you arrived at saturation  at 12 - relative to the paper by Hennink et al

Reply: Thank you. In our methods section we have stated that ‘Sampling: Our convenience sampling strategy aimed to recruits a sample of 12 partici-pants between May 2022 and October 2022. This sample size was estimated to achieve data saturation [11]. No formal saturation checking criteria was set a-priori.

In methodological terms we conducted a ‘nine plus three’ approach where we set the themes/sub-themes after nine interviews and then tested these with three more to judge saturation. We have added one extra bit of information to make this clearer ‘From this, a preliminary coding scheme was produced, and applied, across all nine interviews that had been conducted. Using the preliminary coding scheme all the transcripts were reviewed again with comparisons made between the generated codes and the data to allow the incorporation of consistent and differing responses (RM). Patterns of this coded data were collated into broader concepts which linked them to-gether (i.e. themes) [12-14]. The derivation, review, and refinement of themes were discussed (RM and PF). Each theme was defined and had a clear narrative that was relevant to the aim of this study. The agreed themes and subthemes were then critiqued and refined further (AT and GS). Three further interviews were conducted, reaching the sample size of 12 participants. These were then coded and compared against the thematic structure and recruitment finished at that point, and saturation was judged to have been achieved. The reporting of this study conforms to the consolidated criteria for reporting qualitative studies (COREQ) guidelines [15].

In our limitations section we have stated that ‘no formal saturation stopping criteria were set a-priori, which may have led to a lack of full saturation’ We have added this additional part to this section ‘(i.e. we used judgement, rather than a predefined rule)’.

Reviewer 2: Lines 116 and 117 -Explain the development of the questions relating to the terms in brackets in the interview schedule e.g question 1 (coherence)

Reply: Thank you. We have added one additional sentence ‘Questions were structured around the broad concepts of Normalisation Process Theory’

Reviewer 2: How was the demographic information in table 1 obtained - not included in the interview

Reply: Thank you. We have added new detail to this affect – ‘Demographic characteristics were collected after the recorded interview was complete.’

Reviewer 3 Report

Comments and Suggestions for Authors

This report is a qualitative survey in patients with receiving new service in Scotland.

The manuscript is well-written and well organized.

I would like to recommend a minor revision before publishing.

1. Were there any incentives for participants to attend the survey? Because more than half of the patients were the most deprived, any incentives may affect the selection of participants.

2. Do you have any other background of participants? ie., the length of hospitalization and the number of medications.

3. The conclusions seem to be only general. It is recommended that items specific to the subject of this study be highlighted.

Author Response

Reviewer 3: Were there any incentives for participants to attend the survey? Because more than half of the patients were the most deprived, any incentives may affect the selection of participants.

Reply: Thank you. We have now added the following sentence – ‘No incentives (e.g. financial) were offered for participation.’

Reviewer 3: Do you have any other background of participants? ie., the length of hospitalization and the number of medications.

Reply: No, unfortunately we have no other information on the participants.

Reviewer 3: The conclusions seem to be only general. It is recommended that items specific to the subject of this study be highlighted.

Reply: Thank you. We have slightly reworded this section, in order to show that the issues described come directly from the study results.

‘In this pilot study, patients appreciated that a service where community pharmacy sup-plied their discharge medication ensured a quicker discharge from hospital. Thematic analysis also showed that, stakeholder communication, practical factors (including choice of pharmacy, location of pharmacy, and the practicalities of obtaining the supply from community pharmacy), and a pre-existing relationship and trust in their normal community pharmacy were the key factors that regulated the patient experience. In the main, patients were positive about the introduction of this new ser-vice. However, a lack of previous relationships or trust with a community pharmacy, and previous experiences with medication supply problems were factors which had the potential to negatively impact on patient experiences.’

Reviewer 4 Report

Comments and Suggestions for Authors

This is a well written paper and the results are interesting also for an international audience although the methods show some weakness.

I have few comments.

there is an important bias in the study as the patients were recruited with the very motivational information that participation would ensure a quicker discharge from hospital. I suggest to discuss this limitation.

Data saturation is an issue. The preset number of 12 patients is a very small number. Without proof of data saturation the results and their interpretation is very bad justified. I acknowledge that this issue is mentioned in the limitation, but not in relation to the conclusions. This study should be labelled as a pilot study and further research should follow with data saturation.

Line 351:"so much so that..." typing error ?

Reference 3 should read: https://hospitalpharmacyeurope.com/news/editors-pick/one-stop-dispensing-and-discharge-prescription-time/; I invite the authors to check all URL

Author Response

Reviewer 4: There is an important bias in the study as the patients were recruited with the very motivational information that participation would ensure a quicker discharge from hospital. I suggest to discuss this limitation.

Reply: Thank you. We have added the following statement to the limitations section- ‘At patient recruitment, patients were advised the new service was intended to be quicker, and this may have biased qualitative results.’

Reviewer 4: Data saturation is an issue. The preset number of 12 patients is a very small number. Without proof of data saturation the results and their interpretation is very bad justified. I acknowledge that this issue is mentioned in the limitation, but not in relation to the conclusions.

Reply:  Thank you. In methodological terms we conducted a ‘nine plus three’ approach where we set the themes/sub-themes after nine interviews and then tested these with three more to judge saturation. We have added one extra bit of information to make this clearer ‘From this, a preliminary coding scheme was produced, and applied, across all nine interviews that had been conducted. Using the preliminary coding scheme all the transcripts were reviewed again with comparisons made between the generated codes and the data to allow the incorporation of consistent and differing responses (RM). Patterns of this coded data were collated into broader concepts which linked them to-gether (i.e. themes) [12-14]. The derivation, review, and refinement of themes were discussed (RM and PF). Each theme was defined and had a clear narrative that was relevant to the aim of this study. The agreed themes and subthemes were then critiqued and refined further (AT and GS). Three further interviews were conducted, reaching the sample size of 12 participants. These were then coded and compared against the thematic structure and recruitment finished at that point, and saturation was judged to have been achieved. The reporting of this study conforms to the consolidated criteria for reporting qualitative studies (COREQ) guidelines [15].

In our limitations section we have stated that ‘no formal saturation stopping criteria were set a-priori, which may have led to a lack of full saturation’ We have added this additional part to this section ‘(i.e. we used judgement, rather than a predefined rule)’.

We agree that this is a limitation of the study. We have adapted the following to the Further Research section – ‘A further study with a larger sample and a pre-specified data saturation rule may also give more certainty to the result.’

We also have added this sentence to the conclusion – ‘Further research with a larger sample size and a pre-specified data saturation rule is needed to confirm these results.’

Reviewer 4: This study should be labelled as a pilot study and further research should follow with data saturation.

Reply: Thank You. We have adapted the title as suggested.

Reviewer 4: Line 351:"so much so that..." typing error ?

Reply: Thank you. We have remove ‘So much so that’ and made the sentence easier to read.

Reviewer 4: Reference 3 should read: https://hospitalpharmacyeurope.com/news/editors-pick/one-stop-dispensing-and-discharge-prescription-time/; I invite the authors to check all URL

Reply: Thank you. We have corrected this and a few other URLs.

Round 2

Reviewer 4 Report

Comments and Suggestions for Authors

The Authors have addressed my comments and I have no new one to add. The paper is ready to be published.